Workshop at the 6th Symposium on Advances in Approximate Bayesian Inference (non-archival), 2024 1–32

# On the Properties and Estimation of Pointwise Mutual Information Profiles

**Paweł Czyż**[*]
*ETH AI Center and Department of Biosystems Science and Engineering, ETH Zurich*

**Frederic Grabowski**[*]
*Institute of Fundamental Technological Research, Polish Academy of Sciences*

**Julia E. Vogt**
*Derpartment of Computer Science, ETH Zurich*
*SIB: Swiss Institute of Bioinformatics*

**Niko Beerenwinkel**[†]
*Department of Biosystems Science and Engineering, ETH Zurich*
*SIB: Swiss Institute of Bioinformatics*

**Alexander Marx**[†]
*Research Center Trustworthy Data Science and Security of the University Alliance Ruhr*
*Department of Statistics, TU Dortmund*

## Abstract

Mutual information quantifies the dependence between two random variables. In this work, we explore the pointwise mutual information profile, which is the distribution of the pointwise mutual information values. We analytically describe the profiles of multivariate normal distributions and introduce a novel family of distributions, Bend and Mix Models, for which the profile can be accurately estimated using Monte Carlo methods. We then show how Bend and Mix Models can be used to study the limitations of existing mutual information estimators and understand the effect of experimental outliers on mutual information estimation. Finally, we show how Bend and Mix Models can be used to obtain model-based Bayesian estimates of mutual information, suitable for problems with available domain expertise in which uncertainty quantification is necessary.

## 1. Introduction

Mutual information (MI) is a non-parametric statistical measure used to determine the dependency between two random variables (r.v.s) and numerous approaches have been proposed to estimate mutual information from finite samples (Kay, 1992; Kraskov et al., 2004; Cellucci et al., 2005; Belghazi et al., 2018). Recently, Czyż et al. (2023) proposed an extensive benchmark based on transformed normal and Student distributions and compared the performance of mutual information estimators empirically. They found that different estimators are better suited for particular problems: the KSG estimator (Kraskov et al., 2004) performs well in low- to moderate-dimensional tasks, but is outperformed by neural estimators when dimensionality is sufficiently high or interactions between the r.v.s are sparse. An expected but interesting observation is that a simple model-based estimator

---

[*] Equal contribution.
[†] Joint supervision.

based on canonical correlation analysis (CCA; Kay (1992); Brillinger (2004)) significantly outperforms all other estimators on normal r.v.s. Although model-based estimators have been used in settings with discrete r.v.s (Hutter, 2001; Brillinger, 2004), CCA is the only model-based MI estimator for continuous r.v.s. We consider the development of more advanced model-based MI estimators to be an important open problem.

Additionally, we note a major limitation of the benchmark proposed by Czyż et al. (2023): transforming simple variables cannot yield arbitrary continuous distributions. We prove this by studying the pointwise mutual information profile, a more general functional invariant. To address this problem we introduce a novel class of distributions, *Bend and Mix Models (BMMs)*, which can be efficiently sampled from and allow for reliable estimation of the true MI using Monte Carlo sampling. BMMs include normalizing flows (Kobyzev et al., 2021; Papamakarios et al., 2021) and Gaussian mixtures, and thus can approximate arbitrary distributions. We show how to construct novel benchmark tasks with BMMs in Sec. 3, which fill the gaps in previous benchmarks (Czyż et al., 2023), and allow us to study the robustness of MI estimation to outlier and inlier noise, which is of interest in experimental sciences.

BMMs can also be used for model-based MI estimation, generalizing the existing CCA estimator and estimators for discrete variables of Brillinger (2004); Hutter (2001) and Wu and Yang (2016). Although constructing a BMM estimator requires imposing additional assumptions compared to general-purpose estimators, explicit modeling assumptions can provide useful biases for specific families of distributions, as seen by the success of CCA estimators in the work of Czyż et al. (2023). Moreover, BMMs allow for principled Bayesian inference, which provides sound quantification of epistemic uncertainty in mutual information estimates.

**Contributions** We show that the method of generating tasks used by Czyż et al. (2023) cannot model arbitrary distributions. In particular, we prove that marginal transformations do not change the PMI profile and analytically determine the PMI profile for multivariate normal distributions (Sec. 2). To address this limitation, we introduce the class of Bend and Mix Models (BMMs), for which the true MI can be reliably estimated using Monte Carlo sampling (Sec. 2). We present three distinct applications of Bend and Mix Models. First, we show that this class can be used to approximate arbitrary distributions and thus fill in the gaps in existing benchmarks: in Sec. 3.1 we present examples of novel tasks which could not be obtained by the method of Czyż et al. (2023). Second, we investigate the robustness of mutual information estimators to inliers and outliers (Sec. 3.2). Last, we show how BMMs can be used to estimate MI, together with its epistemic uncertainty, when a suitable parametric model of the generative process is available (Sec. 3.3), generalizing the CCA estimator.

## 2. Theoretical framework

We consider r.v.s valued in smooth manifolds without boundary (Lee, 2012, Ch. 1) equipped with reference measures. For a given pair of smooth manifolds $\mathcal{X}$ and $\mathcal{Y}$, we equip their product $\mathcal{X} \times \mathcal{Y}$ with the product measure and define the set $\mathcal{P}(\mathcal{X}, \mathcal{Y})$ to consist of all probability measures $P_{XY}$ on $\mathcal{X} \times \mathcal{Y}$ such that the joint measure $P_{XY}$, as well as the marginal measures $P_X(A) = P_{XY}(A \times \mathcal{Y})$ and $P_Y(B) = P_{XY}(\mathcal{X} \times B)$, have smooth and

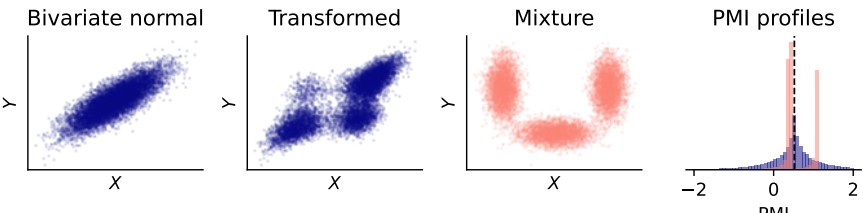

Figure 1: First two panels: a bivariate normal distribution and a transformed distribution sharing the same PMI profile (blue histogram in the fourth panel). Third panel: mixture distribution with distinct PMI profile, which cannot be obtained as a transformation of multivariate normal distribution due to a different PMI profile (pink histogram). All three distributions have the same mutual information, marked with the black line in the fourth panel.

positive PDFs (or PMFs) $p_{XY}$, $p_X$ and $p_Y$ with respect to the reference measures on $\mathcal{X} \times \mathcal{Y}$, $\mathcal{X}$ and $\mathcal{Y}$, respectively. The primary reason for studying $\mathcal{P}(\mathcal{X}, \mathcal{Y})$ is a convenient formula for pointwise mutual information (Pinsker and Feinstein, 1964, Ch. 2) and its profile:

**Definition 1** *Let $X$ and $Y$ be r.v.s valued in $\mathcal{X}$ and $\mathcal{Y}$, respectively, such that $P_{XY} \in \mathcal{P}(\mathcal{X}, \mathcal{Y})$. Pointwise mutual information (PMI) is defined[1] as $\mathrm{PMI}_{XY}(x, y) = \log p_{XY}(x, y) - \log(p_X(x)p_Y(y))$. The PMI profile[2] $\mathrm{Prof}_{XY}$ is the distribution of a r.v. $T = \mathrm{PMI}_{XY}(X, Y)$.*

The mutual information is the first moment of the profile, $\mathbf{I}(X; Y) = \mathbb{E}_{T \sim \mathrm{Prof}_{XY}}[T]$, and is known to be invariant under diffeomorphisms (Kraskov et al., 2004, Appendix). More generally, in Appendix A.1 we show that the whole profile is invariant (see Fig. 1):

**Theorem 2** *Let $P_{XY} \in \mathcal{P}(\mathcal{X}, \mathcal{Y})$ and $f: \mathcal{X} \to \mathcal{X}$ and $g: \mathcal{Y} \to \mathcal{Y}$ be diffeomorphisms. Then for $X' = f(X)$ and $Y' = g(Y)$ it holds that $P_{X'Y'} \in \mathcal{P}(\mathcal{X}, \mathcal{Y})$ and $\mathrm{Prof}_{XY} = \mathrm{Prof}_{X'Y'}$.*

In Appendix A.2 we characterize the profiles of simple distributions, the most important result being:

**Theorem 3** *Let $X$ and $Y$ be r.v.s such that the joint distribution $P_{XY} \in \mathcal{P}(\mathbb{R}^m, \mathbb{R}^n)$ is multivariate normal. If $k = \min(m, n)$ and $\rho_1, \rho_2, \ldots, \rho_k$ are canonical correlations between $X$ and $Y$, then the profile $\mathrm{Prof}_{XY}$ is a generalized $\chi^2$ distribution, namely the distribution of the variable $T = \mathbf{I}(X; Y) + 0.5 \sum_{i=1}^{k} \rho_i (Q_i - Q_i')$, where $Q_i$ and $Q_i'$ are i.i.d. variables sampled according to the $\chi_1^2$ distribution.*

Since the PMI profile is invariant under diffeomorphisms, the approach employed by Czyż et al. (2023) cannot construct distributions with PMI profiles different from the transformed distribution. However, we can obtain distributions with new PMI profiles by using mixtures of distributions (Fig. 1). Hence, we introduce *Bend and Mix Models* which combine

---

1. We use the natural logarithm, meaning that all quantities are measured in nats.
2. Although we are not aware of a prior formal definition and studies of the PMI profile, histograms of approximate PMI between words have been studied before in the computational linguistics community (Allen and Hospedales, 2019).

both tactics: *bending* a distribution (transforming with diffeomorphisms, i.e., normalizing flows) and *mixing* (combining multiple models into a mixture model). Although the mixing operation generally leads to distributions whose PMI profile is not available analytically, we can efficiently construct numerical approximations. To ensure this, we require the following property:

**Definition 4** *Every distribution $P_{XY} \in \mathcal{P}(\mathcal{X}, \mathcal{Y})$ for which we can efficiently sample $(X, Y) \sim P_{XY}$ and numerically evaluate the densities $p_{XY}(x, y)$, $p_X(x)$ and $p_Y(y)$ at every point $(x, y) \in \mathcal{X} \times \mathcal{Y}$ is considered a Bend and Mix Model.*

Any distribution that satisfies this definition can be used as a basic building block and more complex distributions can then be constructed using bending and mixing operations (see Appendix A.4 for further details). The properties of BMMs are chosen so that we can estimate the PMI profile and mutual information with Monte Carlo approaches. Namely, we can sample $T \sim \text{Prof}_{XY}$ by sampling a data point $(x, y)$ and evaluating $t = \text{PMI}_{XY}(x, y)$. Then, MI can be approximated with a Monte Carlo estimate of the integral $\mathbf{I}(X; Y) = \mathbb{E}[T]$. Assuming $\mathbf{I}(X; Y) < \infty$, the Monte Carlo estimator of the mutual information is guaranteed to be unbiased. For a detailed discussion of Monte Carlo standard error (MCSE) under different regularity conditions see Flegal et al. (2008) and Koehler et al. (2009). Analogously, to estimate the PMI profile, we can approximate it with a histogram: for a bin $B \subset \mathbb{R}$ one can introduce its indicator function $\mathbf{1}_B$ and integrate $\mathbb{E}[\mathbf{1}_B(T)]$. Its cumulative density function can be approximated with an empirical sample using the expectations $\mathbb{E}[\mathbf{1}_{(-\infty, a_n]}(T)]$ for a given sequence $(a_n)$. As the characteristic functions are bounded between 0 and 1, the Monte Carlo estimator of both quantities is unbiased and has standard error bounded from above by $1/\sqrt{4n}$ due to the inequality of Popoviciu (1935).

## 3. Case studies

In this section, we apply Bend and Mix Models to three distinct problems. In Sec. 3.1 we demonstrate how they can be used to extend existing benchmarks of mutual information estimators. In Sec. 3.2 we show how BMMs can be used in experimental sciences to investigate the robustness of mutual information to outliers and inliers. In Sec. 3.3 we show how BMMs can be used to provide mutual information estimates in a Bayesian manner.

### 3.1. Novel distributions for estimator evaluation

Czyż et al. (2023) benchmarked mutual information estimators using r.v.s $(X, Y)$ distributed according to multivariate normal and Student distributions for which mutual information is analytically tractable and their transformations $(f(X), g(Y))$, where $f$ and $g$ are chosen so that $\mathbf{I}(f(X); g(Y)) = \mathbf{I}(X; Y))$. However, transforming simple distributions does not ensure that the family of distributions is diverse enough: if $f$ and $g$ were diffeomorphisms, the diversity of the PMI profiles would be very limited due to Theorem 2. The proposed family of BMMs can be used as a more expressive alternative to the multivariate normal and Student distributions.

We implemented four low-dimensional distributions (we visualise samples from the distributions considered in Fig. 2 and defer the detailed description to Appendix C.1), sampled

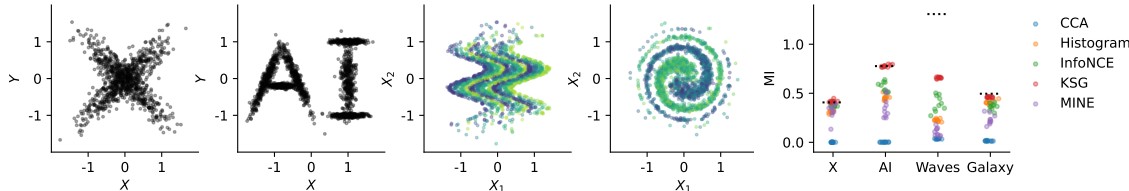

Figure 2: Samples from the proposed distributions. Distributions X and AI represent one-dimensional variables $X$ and $Y$. Distributions Waves and Galaxy plot two-dimensional $X$ variable using spatial coordinates, while one-dimensional $Y$ variable is represented by color. The rightmost plot presents MI estimates compared to the ground-truth (dotted line).

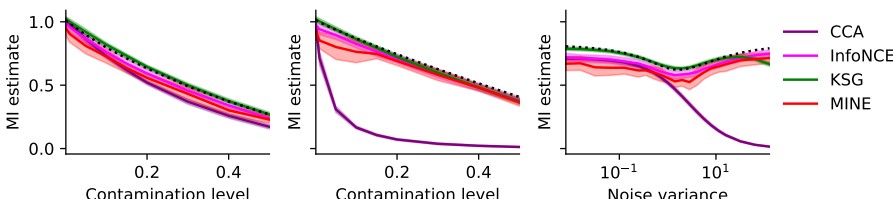

Figure 3: Left: increasing the contamination level $\alpha$ with inlier noise distribution. Middle: increasing the contamination level $\alpha$ with outlier noise distribution. Right: increasing the variance of the noisy normal distribution for constant contamination of 20%. Outliers have less impact than inliers.

ten data sets with $N = 5\,000$ points and applied five estimators: the histogram-based estimator (Cellucci et al., 2005; Darbellay and Vajda, 1999), the popular KSG estimator (Kraskov et al., 2004), canonical correlation analysis (Kay, 1992; Brillinger, 2004) and two neural estimators: InfoNCE (Oord et al., 2018) and MINE (Belghazi et al., 2018) (see Appendix C.2 for hyperparameters used). The estimates are shown in Fig. 2.

Even though the considered problems are low-dimensional and do not encode more information than 1.5 nats, they pose a considerable challenge for the estimators. The KSG estimator, which performs well in low-dimensional tasks, gave the best estimate in all tasks. However, the Waves task was not solved by any estimator. The CCA estimator, excelling at distributions that are close to multivariate normal (Czyż et al., 2023), is not able to capture any information at all. This suggests that Bend and Mix Models can provide a rich set of distributions that can be used to test mutual information estimators.

## 3.2. Modeling outliers

In this section, we use BMMs to study the effect of inliers and outliers on mutual information estimation. Consider an electric circuit or a biological system modeled as a communication channel $p_{Y|X}(y \mid x)$: the researcher controls the input variable $X$ and measures the outcome

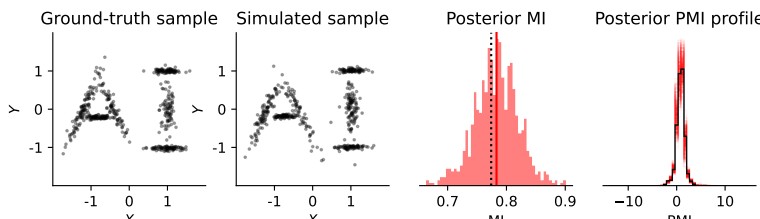

Figure 4: From left to right: data generated according to the AI distribution, data generated according to a single MCMC sample, posterior distribution of the mutual information (red line denotes posterior mean and black line denotes the ground-truth value), and the posterior distribution of the PMI profile (black curve denotes the ground-truth value).

variable $Y$. The mutual information $\mathbf{I}(X;Y)$ is then estimated based on the experimental samples (Nałęcz-Jawecki et al., 2023).

However, every experimental system can suffer from occasional failures. We model the output of a failing system with a noise distribution with a PDF $n(y)$. If the probability of system failure and the distribution of noise $n(y)$ does not depend on the input value $x$, the communication channel becomes a mixture: $p_{Y'|X}(y \mid x) = (1 - \alpha)p_{Y|X}(y \mid x) + \alpha n(y)$.

If the system failure is unnoticed, one can only measure the r.v. $Y'$, rather than $Y$. It is therefore of interest to understand how much $\mathbf{I}(X;Y)$ and $\mathbf{I}(X;Y')$ can differ. In Appendix A.3 we prove that $\mathbf{I}(X;Y') \leq (1 - \alpha)\mathbf{I}(X;Y)$; however, BMMs can evaluate this quantity exactly. We consider a setting with a two-dimensional input variable $X$ and two-dimensional output variables $Y$ and $Y'$. As the joint density $p_{XY}$ we used a multivariate normal with unit scale and correlations $\mathrm{Corr}(X_1,Y_1)=\mathrm{Corr}(X_2,Y_2)=0.8$ and for the noise $n(y)$ we used a multivariate normal distribution with covariance $\sigma^2 I_2$. If $\sigma^2 \approx 1$ this results in inliers, where the noise distribution is hard to distinguish from the signal. For $\sigma^2 \ll 1$ the system failures are all close to 0, while outliers are present for $\sigma^2 \gg 1$.

In Fig. 3 we present the results of three experiments: in the first two, we changed the contamination level $\alpha \in [0, 0.5]$ for $\sigma^2 = 1$ (inlier noise) and $\sigma^2 = 5^2$ (outlier noise) respectively. In the third experiment, we fixed $\alpha = 0.2$ and varied $\sigma^2 \in [2^{-7}, 2^8]$. We see that the inlier noise results in a slightly faster decrease of mutual information, while the outlier noise decreases almost linearly following the upper bound $(1 - \alpha)\mathbf{I}(X;Y)$. Interestingly, in this low-dimensional setting, the KSG, MINE, and InfoNCE estimators reliably estimate the mutual information $\mathbf{I}(X;Y')$, which can significantly differ from $\mathbf{I}(X;Y)$. Although CCA would be the preferred method to estimate $\mathbf{I}(X;Y)$ without any noise (Czyż et al., 2023), even a small number of outliers ($\alpha = 5\%$) can result in unreliable estimates.

### 3.3. Model-based estimation

As the final application of BMMs, we consider the problem of model-based Bayesian estimation of mutual information. Consider a statistical model $\{P_\theta \mid \theta \in \Theta\}$ such that all $P_\theta$ are BMMs and a prior $P(\theta)$. By $\mathbf{I}(P_\theta)$ we will understand $\mathbf{I}(X_\theta;Y_\theta)$, where $(X_\theta, Y_\theta) \sim P_\theta$. Once the data sample is observed, one can apply Bayesian inference algorithms to con-

struct a sample $\theta_1, \ldots, \theta_M$ from the posterior distribution. Although the exact values for $\mathbf{I}(P_{\theta_1}), \ldots, \mathbf{I}(P_{\theta_M})$ are not available, they can be approximated as in Sec. 2. Hence, we can construct an approximate posterior distribution and quantify epistemic uncertainty of the estimate (Fig. 4). Since BMMs include a wide family of distributions, this approach can be used as a general technique for building model-based mutual information estimators, where the generative model can be constructed using domain knowledge. Moreover, this is the first Bayesian estimator of the PMI profile: as all distributions $P_{\theta_m}$ are BMMs, one can construct $M$ histograms (or CDFs) approximating the profile.

To illustrate this approach we implemented a sparse Gaussian mixture model (see Appendix D) to obtain a Bayesian posterior conditioned on 500 data points from the AI distribution (see Fig. 4). We see that the posterior is concentrated around the ground-truth mutual information value and the ground-truth PMI profile is well-approximated by the posterior samples.

## 4. Conclusion

In this article, we have studied pointwise mutual information profiles, determining them analytically for multivariate normal distributions (Theorem 3), and proposed the family of Bend and Mix Models (BMMs), which include multivariate normal and Student distributions, mixture models and normalizing flows, for which the profile can be approximated using Monte Carlo methods. We showed how BMMs can be used to provide novel benchmark tasks to test mutual information estimators, calculate mutual information transmitted through a communication channel in the presence of inliers and outliers, and to provide Bayesian estimates of mutual information between continuous r.v.s. Although this approach is not universal, we find it suitable for problems with precise domain knowledge available and in which uncertainty quantification is desired.

**Acknowledgments**   FG was supported by the OPUS 18 grant operated by Narodowe Centrum Nauki (National Science Centre, Poland) 2019/35/B/NZ2/03898. PC and AM were supported by a fellowship from the ETH AI Center. Part of the work was conducted as AM was at the BSSE department at ETH Zurich. We would like to thank Paweł Nałęcz-Jawecki and Julia Kostin for helpful suggestions on the manuscript.

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

## Appendix A. Technical results

### A.1. Proof of the invariance of the pointwise mutual information profile

Recall a well-known result (Kraskov et al., 2004, Appendix):

**Lemma 5 (Invariance of PMI)** *Let $X' = f(X)$ and $Y' = g(Y)$, where $f$ and $g$ are diffeomorphisms. Then for every $x'$ and $y'$ we have*

$$\mathrm{PMI}_{X'Y'}(x', y') = \mathrm{PMI}_{XY}(x, y),$$

*where $x = f^{-1}(x')$ and $y = g^{-1}(y')$.*

**Proof** From

$$p_{X'Y'}(x', y') = p_{XY}(x, y) \left| \det D\left(f^{-1} \times g^{-1}\right)(x', y') \right|$$

and analogous quantities we conclude that $p_{X'Y'}$ as well as $p_{X'}$ and $p_{Y'}$ are smooth and everywhere positive functions, so that $\mathrm{PMI}_{X'Y'}$ is well-defined. As $D(f^{-1} \times g^{-1})(x', y')$ is a block matrix with $Df^{-1}(x')$ and $Dg^{-1}(y')$ blocks on the diagonal and other blocks zero, we have $\det D(f^{-1} \times g^{-1})(x', y') = \det Df^{-1}(x') \cdot \det Dg^{-1}(y')$. ∎

Now we can prove:

**Theorem 2** *Let $P_{XY} \in \mathcal{P}(\mathcal{X}, \mathcal{Y})$ and $f \colon \mathcal{X} \to \mathcal{X}$ and $g \colon \mathcal{Y} \to \mathcal{Y}$ be diffeomorphisms. Then for $X' = f(X)$ and $Y' = g(Y)$ it holds that $P_{X'Y'} \in \mathcal{P}(\mathcal{X}, \mathcal{Y})$ and $\mathrm{Prof}_{XY} = \mathrm{Prof}_{X'Y'}$.*

**Proof** From the proof of Lemma 5 we conclude that $P_{X'Y'} \in \mathcal{P}(\mathcal{X}, \mathcal{Y})$. Then, we note the profile is the pushforward measure

$$\mathrm{Prof}_{XY} := (\mathrm{PMI}_{XY})_{\#} P_{XY}.$$

Now let $B \subseteq \mathbb{R}$ be any set in the Borel $\sigma$-algebra and $\mathbf{1}_B$ be its characteristic function. Using the change of variables formula for pushforward measure and invariance of PMI:

$$
\begin{aligned}
\operatorname{Prof}_{X'Y'}(B) &= \int \mathbf{1}_B(t) \, \mathrm{d}((\operatorname{PMI}_{X'Y'})_\# P_{X'Y'})(t) \\
&= \int \mathbf{1}_B\left(\operatorname{PMI}_{X'Y'}(x', y')\right) \, \mathrm{d}P_{X'Y'}(x', y') \\
&= \int \mathbf{1}_B\left(\operatorname{PMI}_{X'Y'}(f(x), g(y))\right) \, \mathrm{d}P_{XY}(x, y) \\
&= \int \mathbf{1}_B\left(\operatorname{PMI}_{XY}(x, y)\right) \, \mathrm{d}P_{XY}(x, y) \\
&= \operatorname{Prof}_{XY}(B)
\end{aligned}
$$

$\blacksquare$

## A.2. Pointwise Mutual Information Profiles

The following result shows that the distributions in all $\mathcal{P}(\mathcal{X}, \mathcal{Y})$ classes with zero mutual information have the same profile:

**Proposition 6** *Let $X$ and $Y$ be r.v.s with joint distribution $P_{XY} \in \mathcal{P}(\mathcal{X}, \mathcal{Y})$. Then, $\mathbf{I}(X; Y) = 0$ if and only if $\operatorname{Prof}_{XY} = \delta_0$ is the Dirac measure with a single atom at $0$.*

**Proof** If $\operatorname{Prof}_{XY} = \delta_0$, then the expected value is $\mathbf{I}(X; Y) = 0$. To prove the converse, if $\mathbf{I}(X; Y) = 0$, then $X$ and $Y$ are independent. Hence, $p_{XY}(x, y) = p_X(x)p_Y(y)$ at every point $(x, y)$ and $\operatorname{PMI}_{XY}(x, y) = 0$ everywhere. $\blacksquare$

The following result characterizes the PMI profiles for discrete r.v.s:

**Proposition 7** *If $X$ and $Y$ are discrete r.v.s with $P_{XY} \in \mathcal{P}(\mathcal{X}, \mathcal{Y})$, then the PMI profile is discrete:*

$$
\operatorname{Prof}_{XY} = \sum_{x \in \mathcal{X}} \sum_{y \in \mathcal{Y}} p_{XY}(x, y) \, \delta_{\operatorname{PMI}_{XY}(x,y)}.
$$

**Proof** The measure $P_{XY}$ is discrete and given by

$$
P_{XY} = \sum_{x \in \mathcal{X}} \sum_{y \in \mathcal{Y}} p_{XY}(x, y) \delta_{(x,y)},
$$

so its pushforward by the $\operatorname{PMI}_{XY}$ function has the form

$$
\operatorname{Prof}_{XY} = (\operatorname{PMI}_{XY})_\# P_{XY} = \sum_{x \in \mathcal{X}} \sum_{y \in \mathcal{Y}} p_{XY}(x, y) \delta_{\operatorname{PMI}_{XY}(x,y)}.
$$

$\blacksquare$

The final result characterizes PMI profiles of multivariate normal distributions and is based on the notion of canonical correlations (Jendoubi and Strimmer, 2019).

**Theorem 3** *Let $X$ and $Y$ be r.v.s such that the joint distribution $P_{XY} \in \mathcal{P}(\mathbb{R}^m, \mathbb{R}^n)$ is multivariate normal. If $k = \min(m, n)$ and $\rho_1, \rho_2, \ldots, \rho_k$ are canonical correlations between $X$ and $Y$, then the profile $\mathrm{Prof}_{XY}$ is a generalized $\chi^2$ distribution, namely the distribution of the variable $T = \mathbf{I}(X; Y) + 0.5 \sum_{i=1}^{k} \rho_i (Q_i - Q_i')$, where $Q_i$ and $Q_i'$ are i.i.d. variables sampled according to the $\chi_1^2$ distribution.*

**Proof** Without loss of generality assume that $m \leq n$. As the PMI profile is invariant to diffeomorphisms (Theorem 2), we can also assume that variables $X$ and $Y$ have been whitened by applying canonical correlation analysis (Jendoubi and Strimmer, 2019), that is $\mathbb{E}[X] = 0$, $\mathbb{E}[Y] = 0$ and the covariance matrix is given by

$$\Sigma = \begin{pmatrix} I_m & \Sigma_{XY} \\ \Sigma_{XY}^T & I_n \end{pmatrix} = \begin{pmatrix} I_m & R & 0 \\ R & I_m & 0 \\ 0 & 0 & I_{n-m} \end{pmatrix}$$

where

$$\Sigma_{XY} = \begin{pmatrix} R & 0_{m \times (n-m)} \end{pmatrix}$$

is an $m \times n$ matrix with the last $n - m$ columns being zero vectors and $R = \mathrm{diag}(\rho_1, \ldots, \rho_m)$ being the $m \times m$ diagonal matrix representing canonical correlations.

We will write the inverse in the block form

$$\Sigma^{-1} = \begin{pmatrix} \Lambda_X & \Lambda_{XY} \\ \Lambda_{XY}^T & \Lambda_Y \end{pmatrix} = \begin{pmatrix} \Lambda_X & \tilde{R} & 0 \\ \tilde{R} & \Lambda_X & 0 \\ 0 & 0 & I_{n-m} \end{pmatrix}$$

where the blocks have been calculated using the formula from Petersen and Pedersen (2012, Sec. 9.1):

$$\Lambda_X = (I_m - \Sigma_{XY} \Sigma_{XY}^T)^{-1} = \mathrm{diag}\,(u_1, \ldots, u_m)$$
$$\Lambda_Y = (I_n - \Sigma_{XY}^T \Sigma_{XY}) = \mathrm{diag}\,(u_1, \ldots, u_m, 1, \ldots, 1)$$
$$\Lambda_{XY} = -\Sigma_{XY} \Lambda_Y = \begin{pmatrix} \tilde{R} & 0_{m \times (n-m)} \end{pmatrix},$$

where $\tilde{R} = -\mathrm{diag}\,(u_1 \rho_1, \ldots, u_m \rho_m)$ and $u_i = 1/\left(1 - \rho_i^2\right)$.

We define a quadratic form

$$s(x, y) = x^T \Lambda_X x + y^T \Lambda_Y y + 2 x^T \Lambda_{XY} y$$
$$= \sum_{i=1}^{m} u_i \left(x_i^2 + y_i^2 - 2\rho_i x_i y_i\right) + \sum_{j=m+1}^{n} y_j^2$$

which can be used to calculate log-PDFs:

$$\log p_{XY}(x, y) = -\frac{1}{2} s(x, y) - \frac{1}{2} \log \det \Sigma - \frac{m+n}{2} \log 2\pi,$$
$$\log p_X(x) = -\frac{1}{2} x^T x - \frac{m}{2} \log 2\pi,$$
$$\log p_Y(y) = -\frac{1}{2} y^T y - \frac{n}{2} \log 2\pi.$$

Hence,

$$\text{PMI}_{XY}(x,y) = \frac{x^T x + y^T y - s(x,y)}{2} - \frac{1}{2}\log\det\Sigma.$$

We recognize the last summand as

$$\mathbf{I}(X;Y) = \frac{1}{2}\log\left(\frac{\det I_m \cdot \det I_n}{\det\Sigma}\right) = -\frac{1}{2}\log\det\Sigma.$$

Define quadratic form

$$q(x,y) = 2\big(\text{PMI}_{XY}(x,y) - \mathbf{I}(X;Y)\big) = x^T x + y^T y - s(x,y)$$
$$= \sum_{i=1}^{m}\left((1-u_i)\left(x_i^2 + y_i^2\right) + 2\rho_i u_i x_i y_i\right),$$

which has a corresponding matrix

$$Q = \begin{pmatrix} K & F & 0 \\ F & K & 0 \\ 0 & 0 & 0 \end{pmatrix},$$

where

$$K = \text{diag}(1 - u_1, \ldots, 1 - u_m)$$

and

$$F = \text{diag}(\rho_1 u_1, \ldots, \rho_m u_m).$$

We are interested in the distribution of

$$q(X,Y) = \begin{pmatrix} X^T & Y^T \end{pmatrix} Q \begin{pmatrix} X \\ Y \end{pmatrix},$$

where $(X,Y) \sim \mathcal{N}(0,\Sigma)$.

Imhof (1961) presents a general approach to evaluating the distributions of such quadratic forms. Consider a r.v.

$$Z = \begin{pmatrix} \eta \\ \epsilon \\ \xi \end{pmatrix} \sim \mathcal{N}(0, I_{m+n})$$

which is split into blocks of sizes $m$, $m$ and $n-m$. We will construct a linear transformation $A$ such that

$$\begin{pmatrix} X \\ Y \end{pmatrix} = A \begin{pmatrix} \eta \\ \epsilon \\ \xi \end{pmatrix}.$$

Then, the distribution of $q(X,Y)$ is the distribution of

$$Z^T A^T Q A Z, \qquad Z \sim \mathcal{N}(0, I_{m+n}).$$

We will construct $A$ as

$$A = \begin{pmatrix} P_- & P_+ & 0 \\ -P_- & P_+ & 0 \\ 0 & 0 & I_{n-m} \end{pmatrix}$$

where

$$P_- = \text{diag}\left(\sqrt{\frac{1-\rho_1}{2}}, \cdots, \sqrt{\frac{1-\rho_m}{2}}\right), \qquad P_+ = \text{diag}\left(\sqrt{\frac{1+\rho_1}{2}}, \cdots, \sqrt{\frac{1+\rho_m}{2}}\right).$$

We calculate

$$AA^T = \begin{pmatrix} P_-^2 + P_+^2 & P_+^2 - P_-^2 & 0 \\ P_+^2 - P_-^2 & P_-^2 + P_+^2 & 0 \\ 0 & 0 & I_{n-m} \end{pmatrix} = \begin{pmatrix} I_m & R & 0 \\ R & I_m & 0 \\ 0 & 0 & I_{n-m} \end{pmatrix} = \Sigma$$

and

$$A^T Q A = \begin{pmatrix} 2P_-^2(K-F) & 0 & 0 \\ 0 & 2P_+^2(K+F) & 0 \\ 0 & 0 & 0 \end{pmatrix},$$

where

$$2P_-^2(K-F) = \text{diag}\left(-\rho_1, \ldots, -\rho_m\right), \qquad 2P_+^2(K+F) = \text{diag}\left(\rho_1, \ldots, \rho_m\right).$$

Hence, the distribution of $q(X,Y)$ is the same as the distribution of

$$\sum_{i=1}^m \rho_i(-\eta_i^2 + \epsilon_i^2) + \sum_{j=1}^{n-m} 0 \cdot \xi_j^2,$$

where $(\eta, \epsilon, \xi) \sim \mathcal{N}(0, I_{m+n})$. To summarize, let $Q_1, \ldots, Q_m, Q_1', \ldots, Q_m'$ be i.i.d. random variables distributed according to the $\chi_1^2$ distribution. The quadratic form $q(X,Y)$ has the distribution the same as

$$\sum_{i=1}^m \rho_i(Q_i - Q_i'),$$

which can also be written as

$$q(X,Y) \sim \sum_{i=1}^m \left(\rho_i \chi_1^2 - \rho_i \chi_1^2\right).$$

Note that this distribution is symmetric around 0. We can now reconstruct the profile from $q(X,Y)$:

$$\text{Prof}_{XY} = \mathbf{I}(X;Y) + \sum_{i=1}^m \left(\frac{\rho_i}{2}\chi_1^2 - \frac{\rho_i}{2}\chi_1^2\right),$$

which is symmetric around $\mathbf{I}(X;Y)$ and, in agreement with Proposition 6, degenerates to the atomic distribution $\delta_0$ if and only if $\mathbf{I}(X;Y) = 0$, which is equivalent to $\rho_i = 0$ for all $i$.

As a linear combination of independent $\chi_1^2$ variables, profile has all finite moments. Using the fact that the variance of $\chi_1^2$ distribution is 2, and quadratic scaling of variance, each term has variance $2 \cdot (\rho_i/2)^2 = \rho_i^2/2$. As variances of independent variables are additive, we can sum up all the $2m$ terms to obtain $\rho_1^2 + \cdots + \rho_m^2$. ∎

**Proposition 8** *With fixed mutual information, the variance of the PMI profile is maximized when $\rho_1^2 = \rho_2^2 = \ldots = \rho_m^2$.*

**Proof** Let $a_i = 1 - \rho_i^2$. To maximize the variance we equivalently have to minimize $a_1 + \ldots + a_m$ preserving given constraint on mutual information and $a_i \in (0, 1]$.

The constraint on mutual information takes the form

$$\mathbf{I}(X; Y) = -\frac{1}{2} \sum_{i=1}^{m} \log \left(1 - \rho_i^2\right) = -\frac{1}{2} \log \left(a_1 \cdots a_m\right).$$

Hence, the product $a_1 \cdots a_m$ has to be constant. Denote this constant by $A^m$ for $A \in (0, 1]$ as well.

Let $a_1, \ldots, a_m$ be any minimum of $a_1 + \ldots + a_m$ under the constraints $a_1 \cdots a_m = A^m$ and $a_i \in (0, 1]$. From the inequality between arithmetic and geometric means we note that

$$\frac{a_1 + \ldots + a_m}{m} \geq \sqrt[m]{a_1 \cdots a_m} = A,$$

where the equality holds only if $a_1 = \ldots = a_m = A$. Hence, this is the unique minimum under constraints provided. It follows that $\rho_1^2 = \cdots = \rho_m^2$. Writing $\rho^2$ for the common value, we have

$$\rho^2 = 1 - \exp\left(-2\mathbf{I}(X; Y)/m\right)$$

and

$$V = m\rho^2 = m\left(1 - \exp\left(-2\mathbf{I}(X; Y)/m\right)\right).$$

The mutual information can also be written as function of variance

$$\mathbf{I}(X; Y) = -\frac{1}{2} m \log \left(1 - \rho^2\right) = -\frac{1}{2} m \log(1 - V/m).$$

∎

**Proposition 9** *With fixed non-zero mutual information the variance of the PMI profile is minimized when $\rho_i \neq 0$ for exactly one $i$.*

**Proof** Let $a_i \in (0, 1]$ be any numbers. We have

$$(1 - a_1)(1 - a_2) \geq 0$$

which is equivalent to

$$1 + a_1 a_2 \geq a_1 + a_2$$

where the equality holds if and only if $a_1 = 1$ or $a_2 = 1$.

Using the principle of mathematical induction one can prove a more general inequality:

$$\begin{aligned}
a_1 a_2 \cdots a_m + (m-1) &= 1 + a_1(a_2 \cdots a_m) + (m-2) \\
&\geq a_1 + \left(a_2 \cdots a_m + (m-2)\right) \\
&\geq a_1 + a_2 + \left(a_3 \cdots a_m + (m-3)\right) \\
&\vdots \\
&\geq a_1 + a_2 + \cdots + a_m.
\end{aligned}$$

Let us analyze when the equality can hold. To obtain equality in the first step, we need $a_1 = 1$ or $a_2 \cdots = a_m = 1$, which, given the constraints $a_i \in (0,1]$ would mean that $a_2 = \cdots a_m = 1$. Reasoning inductively, one proves that equality holds only when at least $m-1$ among these numbers are 1.

Let us apply the above reasoning to the numbers $a_i = 1 - \rho_i^2$ and note that we are solving a maximization problem $a_i \in (0,1]$ under a constraint

$$a_1 \cdots a_m = P.$$

Using the argument above we note that all the maxima for the above problem are permutations of the sequence $P, 1, 1, \ldots, 1$. This proves that at most one $\rho_i^2 \neq 0$, what results in at most one $\rho_i \neq 0$. ∎

### A.3. Proof of the failing channel inequality

In this section we prove:

**Proposition 10** *Consider variables $X$, $Y$ and $Y'$, s.t.*

$$p_{XY'}(x,y) = (1-\alpha)p_{XY}(x,y) + \alpha n(y)p_X(x)$$

*with $\alpha \in [0,1]$. Then, $\mathbf{I}(X;Y') \leq (1-\alpha)\,\mathbf{I}(X;Y)$.*

**Proof** Let $Z \sim \text{Bernoulli}(1-\alpha)$ be an auxiliary variable. We have $(X,Y') \mid Z = 1 \sim P_{XY}$ and $(X,Y') \mid Z = 0 \sim P_X \otimes N_Y$.

From the data processing inequality and chain rule we conclude that

$$\mathbf{I}(X;Y') \leq \mathbf{I}(X;Y',Z) = \mathbf{I}(X;Z) + \mathbf{I}(X;Y' \mid Z).$$

Now note that $X$ and $Z$ are independent, so $\mathbf{I}(X;Z) = 0$. Hence,

$$
\begin{aligned}
\mathbf{I}(X;Y') &\leq \mathbf{I}(X;Y' \mid Z) \\
&= \alpha\mathbf{D}_{\text{KL}}\left(P_{XY'|Z=0} \parallel P_{X|Z=0} \otimes P_{Y'|Z=0}\right) + (1-\alpha)\mathbf{D}_{\text{KL}}\left(P_{XY'|Z=1} \parallel P_{X|Z=1} \otimes P_{Y'|Z=1}\right) \\
&= \alpha\mathbf{D}_{\text{KL}}\left(P_{XY'|Z=0} \parallel P_X \otimes N_Y\right) + (1-\alpha)\mathbf{D}_{\text{KL}}\left(P_{XY} \parallel P_X \otimes P_Y\right) \\
&= \alpha\mathbf{D}_{\text{KL}}\left(P_X \otimes N_Y \parallel P_X \otimes N_Y\right) + (1-\alpha)\mathbf{I}(X;Y) \\
&= (1-\alpha)\mathbf{I}(X;Y).
\end{aligned}
$$

∎

### A.4. Constructing new Bend and Mix Models

In this section we prove:

**Proposition 11** *If $P_{XY}$ is a BMM and $f$ and $g$ are diffeomorphisms with numerically tractable Jacobians, then $P_{f(X)g(Y)}$ is a BMM. Moreover, if $P_{X_1Y_1}, \ldots, P_{X_KY_K}$ are BMMs and $w_1, \ldots, w_K$ are positive weights, s.t. $w_1 + \cdots + w_K = 1$, then the mixture distribution $P_{X'Y'} = w_1 P_{X_1Y_1} + \cdots + w_K P_{X_KY_K}$ is a BMM.*

**Proof** From the proof of Lemma 5 and the assumption of tractability of the Jacobians of $f$ and $g$ we obtain the tractability of the formulae for the densities $p_{f(X)g(Y)}$, $p_{f(X)}$ and $p_{g(Y)}$. Sampling $(f(X), g(Y))$ amounts to sampling $(X, Y)$ and then transforming the sample using $f$ and $g$.

To prove that mixtures are fine, note that we can evaluate the densities $p_{X'Y'}(x, y)$, $p_{X'}(x)$ and $p_{Y'}(y)$ of the mixture distribution using the weighted sums of $p_{X_k Y_k}(x, y)$, $p_{X_k}(x)$ and $p_{Y_k}(y)$, respectively. To sample $(X', Y') \sim P_{X'Y'}$ from the mixture distribution we can sample an auxiliary variable $Z \sim \text{Categorical}(K; w_1, \ldots, w_K)$. Then, we have $\big((X', Y') \mid Z = k\big) = (X_k, Y_k)$, so that we can sample from $P_{X_k Y_k}$. ∎

### A.5. Approximation with smooth densities

In this section, we prove the following result:

**Theorem 12** *Let $X$ and $Y$ be random variables with joint distribution $P_{XY}$ and finite mutual information $\mathbf{I}(X; Y) < \infty$. Then, there exists a sequence of random variables $(X_k, Y_k)$ for $k = 1, 2, \ldots$ such that:*

1. *Each $P_{X_k Y_k} \in \mathcal{P}(m, n)$.*

2. *Distributions $P_{X_k Y_k}$ weakly converge to $P_{XY}$ as $k \to \infty$.*

3. *Mutual information is preserved:*

$$\lim_{k \to \infty} \mathbf{I}(X_k; Y_k) = \mathbf{I}(X; Y).$$

The proof will automatically follow from two lemmata.

**Lemma 13** *Let $X$ be a random variable on $\mathbb{R}^k$. Let $N$ be a r.v. independent of $X$ distributed according to the multivariate normal distribution $\mathcal{N}(0, \sigma^2 I)$ with $\sigma > 0$. Then the distribution of $X + N$ has a PDF which is everywhere smooth and positive.*

**Proof** Let $P_X$ be the distribution of $X$ and $f$ be the PDF of the multivariate normal distribution $\mathcal{N}(0, \sigma^2 I)$, which is bounded, smooth, and everywhere positive. The convolution of $P_X$ and normal density has a PDF

$$(f * P_X)(x) = \int f(x - y) \, \mathrm{d}P_X(y) = \mathbb{E}[f(x - X)].$$

From the dominated convergence theorem it follows that all the partial derivatives of $\partial_j (f * P_X)$ exist and are given by $(\partial_j f) * P_X$. From the principle of mathematical induction we conclude that $f * P_X$ is smooth.

Positivity follows from the fact that the random variable $f(x - X)$ is strictly positive, so that its expected value is strictly positive as well. ∎

The second lemma is the following:

**Lemma 14** *Assume that $I(X;Y) < \infty$. Let $X_k = X + N_k$ and $Y_k = Y + M_k$ where $N_k$ and $M_k$ are independent noise variables distributed according to $\mathcal{N}(0, k^{-2}I)$. Then*

$$\lim_{k\to\infty} \mathbf{I}(X_k; Y_k) = \mathbf{I}(X; Y).$$

**Proof** Observe that for every $k$ the equality $\mathbf{I}(X_k; Y_k) \leq \mathbf{I}(X; Y)$ holds. This is a simple corollary of the data processing inequality applied to the variables $(X_k, N_k)$ and $(Y_k, M_k)$. Then, note that $P_{X_k Y_k}$ converges weakly to $P_{XY}$. As mutual information is assumed to be finite, it is given by the Kullback–Leibler divergence, which is lower semicontinuous. This proves the reverse inequality. ∎

### A.6. Creating and destroying information with mixtures

Let $A = (0, 1)$ and $B = (1, 2)$ be two disjoint intervals of unit length. We define two pairs of random variables:

$$(X_1, Y_1) \sim \mathrm{Uniform}(A \times A), \qquad (X_2, Y_2) \sim \mathrm{Uniform}(B \times B).$$

Note that

$$\mathbf{I}(X_1; Y_1) = \mathbf{I}(X_2; Y_2) = 0.$$

If $(X, Y) \sim 0.5 P_{X_1 Y_1} + 0.5 P_{X_2 Y_2}$ is distributed according to a mixture, we have

$$p_{XY}(x, y) = \frac{1}{2}\mathbf{1}[(x, y) \in A \times A \cup B \times B]$$

and

$$p_X(x) = \frac{1}{2}\mathbf{1}[x \in A \cup B], \qquad p_Y(y) = \frac{1}{2}\mathbf{1}[x \in A \cup B].$$

Hence,

$$\mathrm{PMI}_{XY}(x, y) = \log 2 \cdot \mathbf{1}[(x, y) \in A \times A \cup B \times B]$$

and

$$\mathbf{I}(X; Y) = \frac{1}{2}\log 2 + \frac{1}{2}\log 2 = \log 2.$$

For a second example, demonstrating vanishing mutual information, recall the distribution constructed as above:

$$p_{XY}(x, y) = \frac{1}{2}\mathbf{1}[(x, y) \in A \times A \cup B \times B]$$

and a symmetric one

$$p_{UV}(x, y) = \frac{1}{2}\mathbf{1}[(x, y) \in A \times B \cup B \times A].$$

We have

$$\mathbf{I}(X; Y) = \mathbf{I}(U; V) = \log 2.$$

On the other hand, the mixture distribution

$$(Z, T) \sim 0.5 P_{XY} + 0.5 P_{UV}$$

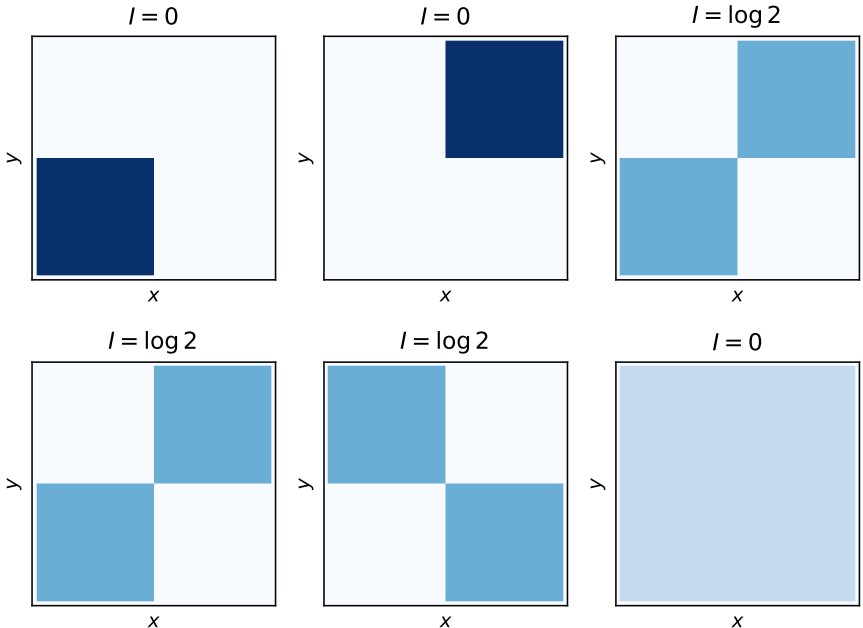

Figure 5: Top row: two distributions with zero MI can be mixed to obtain a distribution with non-zero MI. Bottom row: two distributions with non-zero MI can be mixed to obtain a distribution with zero MI.

has zero mutual information, $\mathbf{I}(Z;T) = 0$, as

$$
\begin{aligned}
p_{ZT}(x,y) &= \frac{1}{4}\mathbf{1}[(x,y) \in (A \cup B) \times (A \cup B)] \\
&= \frac{1}{2}\mathbf{1}[x \in A \cup B] \cdot \frac{1}{2}\mathbf{1}[y \in A \cup B] \\
&= p_Z(x) \cdot p_T(y).
\end{aligned}
$$

Note that similar examples can be constructed using BMMs by using mixtures of multivariate normal distributions.

This demonstrates that mixtures can create and destroy information. There is however an upper bound on the amount of information mixtures can create (see Haussler and Opper (1997) and Kolchinsky and Tracey (2017)):

**Proposition 15** *Consider r.v.s $(X_k, Y_k)$ such that $\mathbf{I}(X_k; Y_k) < \infty$ for $k = 1, \ldots, K$. Let $(X', Y')$ be their mixture with weights $w_1, \ldots, w_K$. Then,*

$$
0 \leq \mathbf{I}(X'; Y') \leq \log K + \sum_{k=1}^{K} w_k \, \mathbf{I}(X_k; Y_k).
$$

*Moreover, these inequalities are tight:*

1. *There exists a mixture such that* $\mathbf{I}(X'; Y') = \log K$ *even though* $\mathbf{I}(X_k; Y_k) = 0$ *for all* $k$.
2. *There exists a mixture such that* $\mathbf{I}(X'; Y') = 0$ *even though* $\mathbf{I}(X_k; Y_k) > 0$ *for all* $k$.

**Proof** The examples provide explicit constructions of distributions with the specified properties. To prove the upper bound, consider a variable $Z \sim \text{Categorical}(K; w_1, \ldots, w_K)$. The random variables corresponding to the mixture distribution, $(X', Y')$, have conditional distributions

$$(X', Y') \mid Z = k \sim P_{X_k Y_k}.$$

From the data processing inequality and chain rule we have

$$\mathbf{I}(X'; Y') \leq \mathbf{I}(X'; Y', Z) = \mathbf{I}(X'; Z) + \mathbf{I}(X'; Y' \mid Z).$$

As $Z$ is discrete, the first summand, $\mathbf{I}(X'; Z)$, is bounded from above by the entropy $H(Z)$ (Polyanskiy and Wu, 2022, Th. 3.4(e)), which cannot exceed $\log K$ (Polyanskiy and Wu, 2022, Th. 1.4(b)). The second summand can be written as

$$\mathbf{I}(X'; Y' \mid Z) = \sum_{k=1}^{K} P(Z = k) \, \mathbf{D}_{\text{KL}} \left( P_{X'Y'|Z=k} \, \| \, P_{X'|Z=k} \otimes P_{Y'|Z=k} \right)$$
$$= \sum_{k=1}^{K} w_k \, \mathbf{D}_{\text{KL}} \left( P_{X_k Y_k} \, \| \, P_{X_k} \otimes P_{Y_k} \right) = \sum_{k=1}^{K} w_k \, \mathbf{I}(X_k; Y_k).$$

$\blacksquare$

## Appendix B. Distributions involving discrete variables

The formalism in Section 2 is applicable to both continuous and discrete random variables, although in Section 3.1 we focus on the distributions in which both $X$ and $Y$ are continuous. If $X$ and $Y$ are discrete, mutual information $\mathbf{I}(X; Y)$ can be calculated analytically from the joint probability matrix and there exist numerous approaches to estimate it from collected samples (Hutter, 2001; Brillinger, 2004).

In this section we consider the mixed case, in which one variable is continuous and the other one is discrete. For example, Carrara and Ernst (2023) describe a particle physics experiment in which $X$ is an 18-dimensional random variable, but $Y$ is binary. Grabowski et al. (2019) consider a cell transmitting information through the MAPK signalling pathway, assuming the input signal $X$ to be discrete and the measured response $Y$ to be continuous.

Yet, there are only a few distributions $P_{XY}$ with known ground-truth mutual information assuming this discrete-continuous case. Gao et al. (2017, Sec. 5) describe a discrete random variable $X$ which is uniformly sampled from the set $\mathcal{X} = \{0, \ldots, m-1\}$ and the continuous $Y$ variable is sampled as

$$(Y \mid X = x) \sim \text{Uniform}(x, x + 2),$$

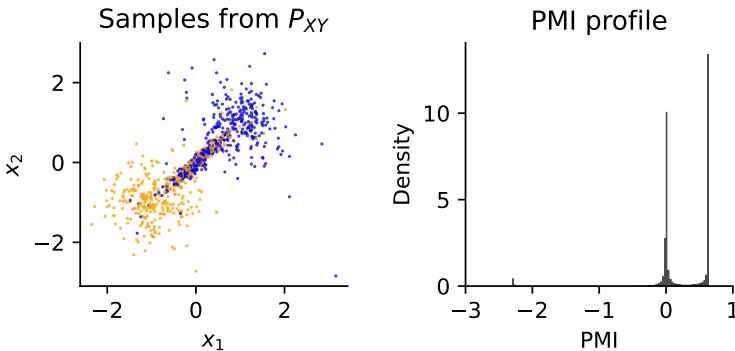

Figure 6: Left: samples from the $P_{XY}$ distribution, colored by the value of the binary variable $Y$. Right: the PMI profile of $P_{XY}$ distribution.

which is therefore distributed on $\mathcal{Y} = (0, m + 1)$. Gao et al. (2017) prove that mutual information in this case is

$$\mathbf{I}(X; Y) = \log m - \frac{m}{m - 1} \log 2$$

for $m \geq 2$ and $\mathbf{I}(X; Y) = 0$ for $m = 1$ and consider a multivariate analogue of this distribution, in which pairs of variables $(X_k, Y_k)$ for $k = 1, \ldots, K$ are sampled independently using the above procedure. Then, they are concatenated into multivariate vectors $(X_1, \ldots, X_K)$ and $(Y_1, \ldots, Y_K)$ with $K$ times larger mutual information.

We will show how to relate the bivariate example to the framework of Bend and Mix Models (multivariate case can be constructed analogously). Note that the joint distribution $P_{XY}$ is not strictly in $\mathcal{P}(\mathcal{X}, \mathcal{Y})$, as it is supported on the one-dimensional manifold $\mathcal{M} \subset \mathcal{X} \times \mathcal{Y}$ with $m$ connected components (cf. Politis (1991) and Marx et al. (2021)), on which

$$p_{XY}(x, y) = \frac{1}{2m} \mathbf{1}[y \in (x, x + 2)],$$

but it is possible to extend the definitions of Section 2, so that this technical difficulty is resolved. The marginal distributions are tractable and admit PDFs:

$$p_X(x) = 1/m,$$
$$p_Y(y) = \sum_{x=0}^{m-1} p_{XY}(x, y).$$

Although $p_Y$ is not smooth on $\mathcal{Y}$ (at integer points), this technical difficulty can also be resolved as this set is of measure zero. Hence, although this distribution is not strictly a BMM, it can still be modelled using the introduced framework.

Next, Gao et al. (2017, Sec. 5) consider the zero-inflated Poissonization of the exponential random variable valued in $\mathcal{X} = \{x \in \mathbb{R} \mid x \geq 0\}$, with: $X \sim \text{Exp}(1)$ and $Y$ being a discrete random variable valued in the set of non-negative integers $\mathcal{Y} = \{0, 1, 2, \ldots\}$:

$$(Y \mid X = x) \sim p\,\delta_0 + (1 - p)\,\text{Poisson}(x).$$

They show that

$$\mathbf{I}(X;Y) = (1-p)\left(2\log 2 - \gamma - \sum_{k=0}^{\infty} \log k \cdot 2^{-k}\right),$$

where $\gamma$ is the Euler–Mascheroni constant.

To use BMMs in this case, one has to formally extend the definition of $\mathcal{P}(\mathcal{X},\mathcal{Y})$ as $\mathcal{X}$ is a manifold with boundary (Lee, 2012, Ch. 1). The joint probability distribution is then given by

$$p_{XY}(x,y) = p_X(x) \cdot p_{Y|X}(y \mid x) = e^{-x} \cdot \left(p \cdot \mathbf{1}[y=0] + (1-p)\frac{e^{-x}x^y}{y!}\right)$$

and the PMF function of the $Y$ variable is also analytically known:

$$p_Y(y) = p \cdot \mathbf{1}[y=0] + (1-p) \cdot 2^{-(1+y)}$$

Hence, the framework of BMMs, with minor technical adjustments, can accommodate the above distributions.

However, it also allows one to create more expressive distributions, for which analytical formulae for ground-truth mutual information are not available, but can be approximated with the Monte Carlo methods as explained in Section 2: consider a continuous random variable $X$ and a discrete random variable $Y$. To introduce a dependency between $X$ and $Y$ variable, we can use a mixture of distributions in which the component variables are independent, i.e., $P_{X_k Y_k} = P_{X_k} \otimes P_{Y_k}$. Therefore, we consider a graphical model $X \leftarrow Z \rightarrow Y$, in which the distributions $P_{X_k} = P_{X|Z=k}$ are known and have tractable PDFs. The distributions of $P_{Y_k} = P_{Y|Z_k}$ are given by probability tables. Monte Carlo estimators of Section 2 can then be used to estimate $\mathbf{I}(X;Y)$ with high accuracy. Note also that this general procedure includes the case $X \leftarrow Y$ by setting $Z = Y$.

We illustrate it in a simple example with $\mathcal{X} = \mathbb{R}^2$, $\mathcal{Y} = \{0,1\}$ and $K = 3$ components.

The first component will model a cluster in the $\mathcal{X}$ space, strongly associated with $Y = 1$ value. For $P_{X_1}$ we use a bivariate Student distribution centered at $(1,1)$ with isotropic dispersion $\Omega = 0.2 \cdot I_2$ and 8 degrees of freedom. We take $P_{Y_1}$ to be the Bernoulli variable with probability $P(Y_1 = 1) = 0.95$.

Analogously, we define a second cluster, strongly associated with $Y = 0$ value: $P_{X_2}$ is a bivariate Student distribution with the same dispersion matrix, but centered at $(-1,-1)$ and with 8 degrees of freedom. Then, $Y_2$ is a Bernoulli variable with $P(Y_2 = 1) = 0.05$.

We then define a third component using a bivariate normal distribution centered at $(0,0)$ and with covariance matrix

$$\Sigma = 0.1 \begin{pmatrix} 1 & 0.95 \\ 0.95 & 1 \end{pmatrix}.$$

This component is not informative of $Y$, that is $P(Y_3 = 1) = 0.5$.

We used weights $w_1 = w_2 = 1/4$ and $w_3 = 1/2$, what resulted in the distribution visualised in Fig. 6. We estimated both the profile and the mutual information using $N = 10^6$ samples and obtained $\mathbf{I}(X;Y) = 0.224$ with MCSE of $5.1 \cdot 10^{-4}$.

In principle, the above construction can be used to generate realistic high-dimensional data sets (e.g., audio or image) with known ground-truth mutual information, by assuming

the generative model $Y \to X$ (i.e., $Z = Y$) and modelling each $P_{X_k}$ using a normalizing flow or an autoregressive model (Murphy, 2023, Ch. 22) trained on an auxiliary data set with fixed label $Y_k = y_k$. Hence, at least in principle, one could obtain highly-expressive generative process $p_{X|Y}(x \mid y)$ with tractable probability and sampling. Pairing this with an arbitrary probability vector $p_Y(y)$ one can obtain $p_{XY}(x, y)$ and $p_X(x)$ even for high-dimensional data sets, so that Monte Carlo estimator can be used to determine the ground-truth mutual information. However, we anticipate possible practical difficulties with scaling up the proposed approach to high-dimensional data and we leave empirical investigation of this topic to future work.

## Appendix C. Experimental details

### C.1. New distributions

All distributions have been implemented in TensorFlow Probability on JAX (Dillon et al., 2017; Bradbury et al., 2018). In this section we assume that $X$ and $Y$ are valued in Euclidean spaces. In Appendix B, we consider cases involving discrete variables.

We constructed the X distribution as a mixture of bivariate normal distributions with equal weights, zero mean and covariance matrices specified by

$$\Sigma_\pm = 0.3 \begin{pmatrix} 1 & \pm 0.9 \\ \pm 0.9 & 1 \end{pmatrix}.$$

Note that the marginal distributions of each of component distributions is $\mathcal{N}(0, 0.3^2)$ and subsequently their mixture has exactly the same marginal distributions. This is therefore an interesting example of a distribution in which the joint probability distribution is not multivariate normal, although the marginal distributions of $X$ and $Y$ variables are normal individually.

The AI distribution was constructed as an equally-weighted mixture of six bivariate normal distributions with equal weights and the following parameters:

$$\mu_1 = (1, 0)$$
$$\Sigma_1 = \mathrm{diag}(0.01, 0.2)$$
$$\mu_2 = (1, 1)$$
$$\Sigma_2 = \mathrm{diag}(0.05, 0.001)$$
$$\mu_3 = (1, -1)$$
$$\Sigma_3 = \mathrm{diag}(0.05, 0.001)$$
$$\mu_4 = (-0.8, -0.2)$$
$$\Sigma_4 = \mathrm{diag}(0.03, 0.001)$$
$$\mu_5 = (-1.2, 0)$$
$$\Sigma_5 = \begin{pmatrix} 0.04 & 0.085 \\ 0.085 & 0.2 \end{pmatrix}$$
$$\mu_6 = (-0.4, 0)$$
$$\Sigma_6 = \begin{pmatrix} 0.04 & -0.085 \\ -0.085 & 0.2 \end{pmatrix}$$

The Galaxy distribution was constructed as an equally-weighted mixture of isotropic multivariate normal distributions with $\mu_{\pm} = \pm(1, 1, 1)$ and unit covariance matrix and the $X$ variable was transformed using the spiral diffeomorphism with $v = 0.5$ (cf. Czyż et al. (2023)).

The Waves distribution was created as an equally-weighted mixture of 12 multivariate normal distributions with equal covariance matrices $\Sigma = \mathrm{diag}(0.1, 1, 0.1)$ and mean vectors

$$\mu_i = (x, 0, x \bmod 4), \quad i \in \{0, 1, \ldots, 11\}.$$

This construction results in a distribution where different vertical components of the $X$ variable are assigned $Y$ values calculated modulo 4. Then, we transformed the $X$ variable with a continuous injection

$$f(x_1, x_2) = (x_1 + 5 \sin(3x_2), x_2),$$

which does not change the mutual information. Finally, we applied the affine mappings

$$a_1(x) = 0.1x - 0.8, \quad a_2(y) = 0.5y,$$

to make the range of the typical values comparable with other distributions.

To estimate the ground-truth mutual information we used the Monte Carlo approach described in Section 2 with $N = 200\,000$ samples.

### C.2. Estimator hyperparameters

Czyż et al. (2023, Appendix E.4) study the effects of hyperparameters on mutual information estimators. We decided to use the histogram-based estimator (Cellucci et al., 2005; Darbellay and Vajda, 1999) with a fixed number of 10 bins per dimension and the popular KSG estimator (Kraskov et al., 2004) with $k = 10$ neighbors. Canonical correlation analysis (Kay, 1992; Brillinger, 2004) does not have any hyperparameters. Finally, we employed neural estimators (InfoNCE (Oord et al., 2018) and MINE (Belghazi et al., 2018)) with the neural critic being a ReLU network with 16 and 8 hidden neurons, as it obtained competitive performance in the benchmark of Czyż et al. (2023, Appendix E.4).

As a preprocessing strategy, we followed Czyż et al. (2023, Appendix E.3) and transformed all samples to have zero empirical mean and unit variance along each dimension.

## Appendix D. Gaussian Mixture Models

Recall from Sec. 3.3 that Bayesian estimation of mutual information consists of the following steps:

1. Propose a parametric generative model of the data, $P_\theta := P(X, Y \mid \theta)$, and assume a prior $P(\theta)$ on the parameter space.

2. Use a Markov chain Monte Carlo method to obtain a sample $\theta^{(1)}, \ldots, \theta^{(m)}$ from the posterior $P(\theta \mid X_1, Y_1, \ldots, X_N, Y_N)$.

3. Estimate mutual information (and the PMI profile) for each $\theta^{(m)}$ using the Monte Carlo method described in Sec. 2.

4. Validate the findings using e.g., posterior predictive checks and cross-validation.

We consider the following sparse Gaussian mixture model with $K = 10$ components:

$$\pi \sim \text{Dirichlet}(K; 1/K, 1/K, \ldots, 1/K),$$
$$Z_n \mid \pi \sim \text{Categorical}(\pi), \qquad n = 1, \ldots, N,$$
$$\mu_k \sim \mathcal{N}\left(0, 3^2 I_D\right), \qquad k = 1, \ldots, K,$$
$$\Sigma_k \sim \text{ScaledLKJ}(1, 1), \qquad k = 1, \ldots, K,$$
$$(X_n, Y_n) \mid Z_n, \{\mu_k, \Sigma_k\} \sim \mathcal{N}(\mu_{Z_n}, \Sigma_{Z_n}), \qquad n = 1, \ldots, N.$$

Sampling a single covariance matrix $\Sigma$ from $\text{ScaledLKJ}(\sigma, \eta)$ distribution corresponds to sampling the correlation matrix $R$ from the Lewandowski-Kurowicka-Joe (LKJ) distribution (Lewandowski et al., 2009):

$$p(R) \propto (\det R)^{\eta - 1},$$

sampling the scale parameters

$$\lambda_1, \lambda_2, \ldots, \lambda_D \sim \text{HalfCauchy}(\text{scale}=\sigma),$$

and then constructing the covariance matrix as $\Sigma_{ij} = R_{ij}\lambda_i\lambda_j$.

The sparse Dirichlet prior is a finite-dimensional alternative to the Dirichlet process, which truncates the number of occupied clusters depending on the data (Frühwirth-Schnatter and Malsiner-Walli, 2019). In particular, the *a priori* expected number of clusters depends on the number of data points to be observed.

To perform Markov chain Monte Carlo sampling, we implemented the model in NumPyro (Phan et al., 2019), with local latent variables $Z_n$ marginalized out and performed sampling using the NUTS sampler (Hoffman and Gelman, 2014). We used 2000 warm-up steps and collected 800 samples.

The $m$th sample is therefore given by

$$\theta^{(m)} = \left(\pi^{(m)}, \left(\mu_k^{(m)}, \Sigma_k^{(m)}\right)_{k=1,\ldots,K}\right)$$

which is then used to parametrize a Gaussian mixture distribution $P_{\theta^{(m)}}$.

Finally, using Monte Carlo method described in Sec. 2 (with 100,000 samples) we then estimated mutual information $\mathbf{I}(P_{\theta^{(m)}})$ and the profile.

We used the above procedure to perform inference in the model applied to the X, AI, Waves and Galaxy distributions, changing the number of data points $N \in \{125, 250, 500, 1000\}$. We visualise the observed sample, a single posterior predictive sample and posterior on mutual information and the PMI profile in Fig. 7, Fig. 8, Fig. 9 and Fig. 10.

We can see that the performance of this method relies on an appropriate model of the true data-generating distribution. As observed in Sec. 3.1, the CCA estimator can yield unreliable estimates when the model is misspecified, that is, the true data-generating process $P_{XY}$ does not belong to the assumed multivariate normal family $P_\theta$ (Watson and Holmes, 2016). To illustrate the risks of using misspecified models, we applied the same Gaussian mixture model to 500 points from the Galaxy distribution (Fig. 4). The misspecification in

this case can be diagnosed via posterior predictive checking (Gelman et al., 2013, Ch. 6): in Fig. 4 we see that a data sample simulated from the model looks substantially different from the observed data, meaning that the model did not capture the distribution well (cf. Appendix D). This provides a clear indication that the estimates should not be trusted: most of the probability mass of the Bayesian posterior is far from the ground-truth mutual information. Similarly, the posterior on the PMI profile is biased. We therefore recommend using extensive model-checking techniques discussed by Sankaran and Holmes (2023, Sec. 4) and Piironen and Vehtari (2017). Alternative approaches to inference with misspecified Bayesian models, as proposed by Grünwald and van Ommen (2017) and Lyddon et al. (2018), may also be applicable in this case, although we leave systematic investigation of their properties to future work.

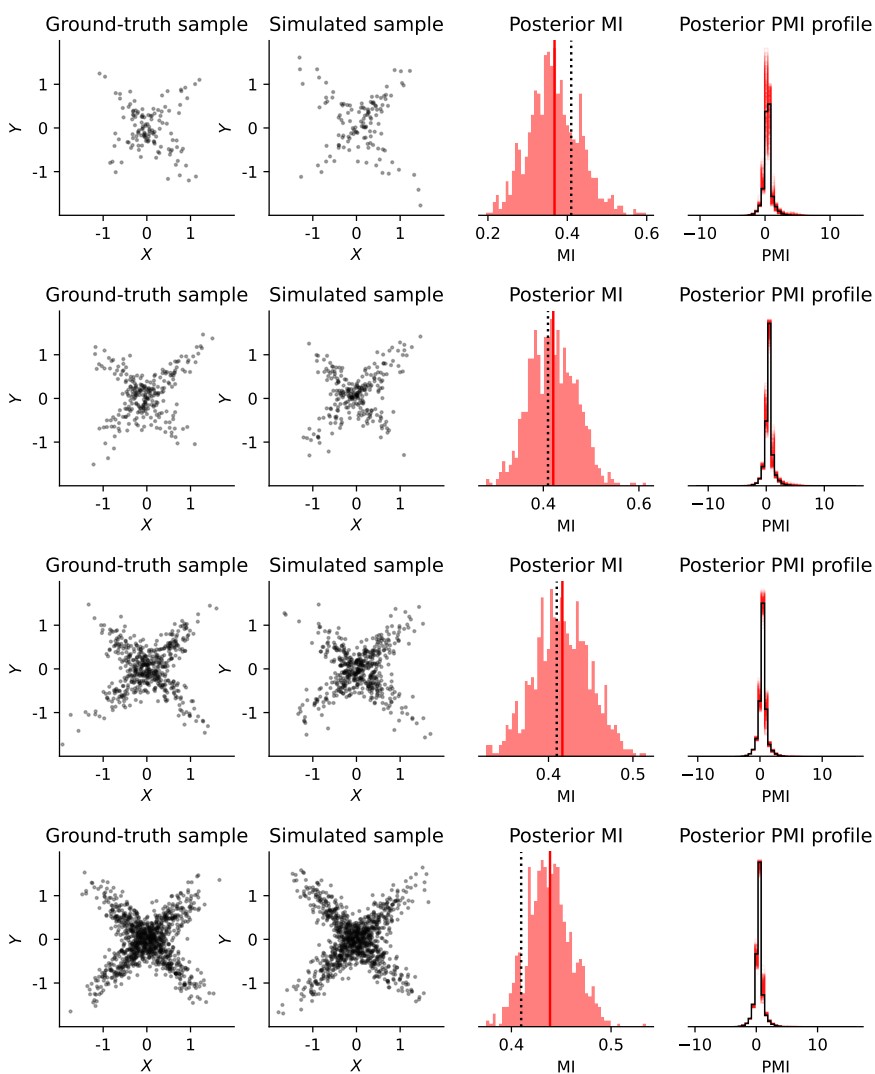

Figure 7: Gaussian mixture model fitted to the X distribution with 125, 250, 500 and 1000 samples.

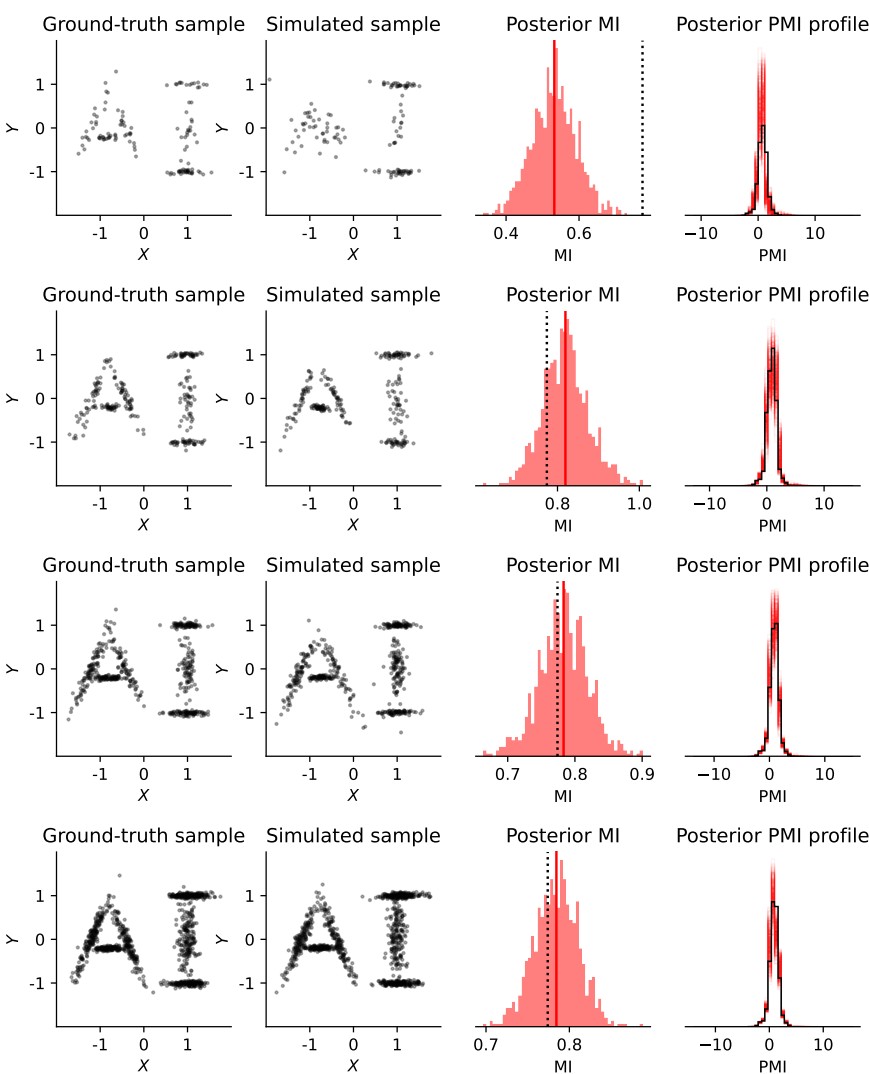

Figure 8: Gaussian mixture model fitted to the AI distribution with 125, 250, 500 and 1000 samples.

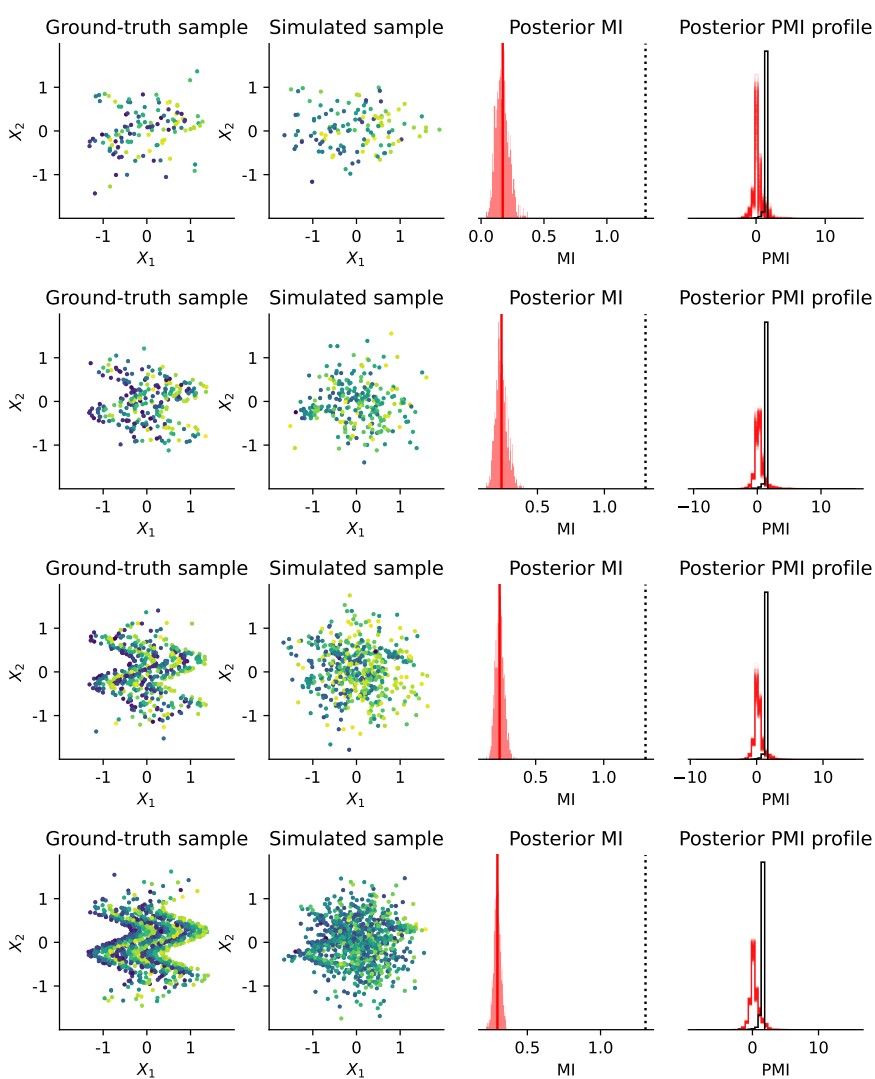

Figure 9: Gaussian mixture model fitted to the Waves distribution with 125, 250, 500 and 1000 samples.

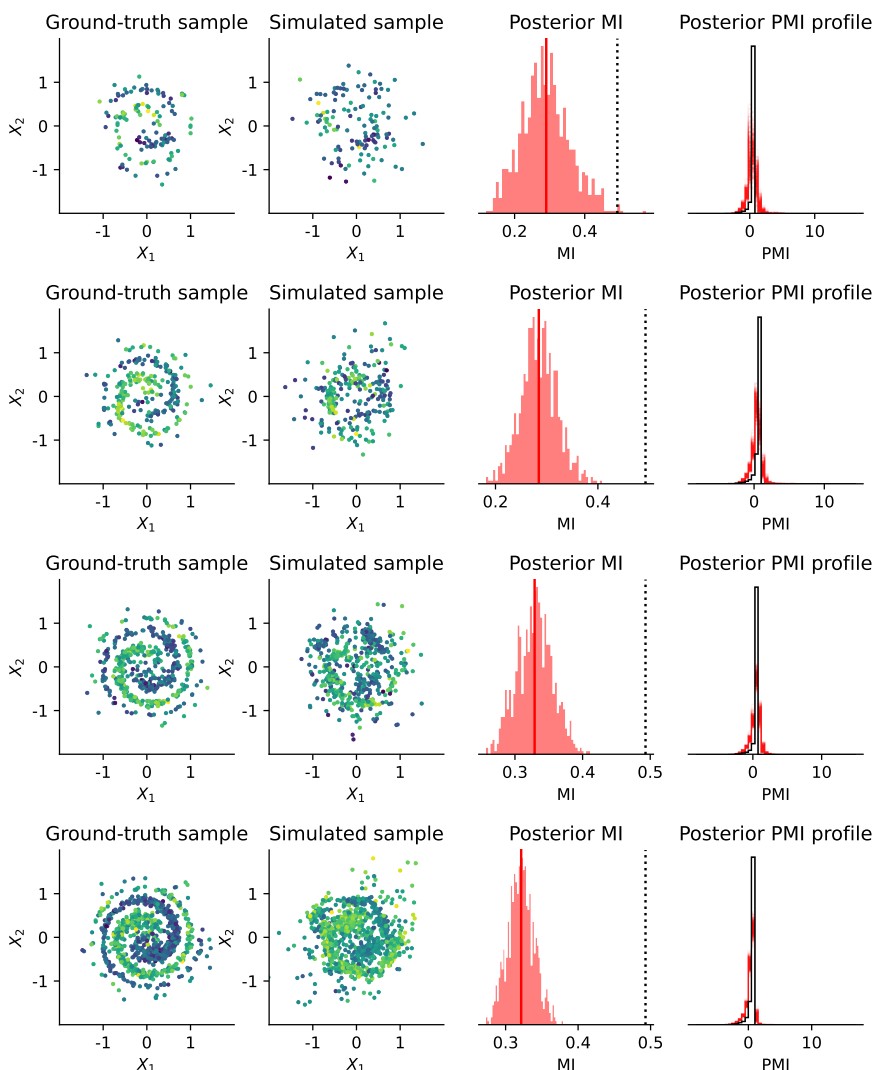

Figure 10: Gaussian mixture model fitted to the Galaxy distribution with 125, 250, 500 and 1000 samples.

