# OpenReview forum: "On the Properties and Estimation of Pointwise Mutual Information Profiles"
_approximateinference.org/AABI/2024/Symposium — AABI 2024_

### Official Review · Reviewer_afzA · 2024-04-17
**Potentially useful benchmark**

**Rating:** 7
**Confidence:** 3

**Review:**

Summary:

After a brief review of the state-of-the-art in estimation of pointwise mutual information profiles (PMI), the authors point out limitations in the generating tasks proposed by Czyz et al. (2023) used as a benchmark to compare these estimators. The authors then propose a new set of generating tasks: Bend and Mixture Models (BMM). They are based on both "bending" a simple distribution (via a diffeomorphism) and mixing various distributions (mixture models). Because   the PMI of mixture of Gaussian distributions is known, the PMI of BMM's obtained by "bending" the Gaussian distributions can be computed explicitly, which offers a good framework to test estimatods. The authors compare the popular estimators on this tasks, and obtain a wide range of results -- including some tasks that are not solved by any of the known methods. This shows that their benchmark is relevant to compare estimators.

Strengths:

- I believe that new benchmarks are very useful for the community.
- The paper is easy to follow. I appendix is well written, with clear and easy-to-follow proofs (note that I did not have time to check ALL the proofs).

Weaknesses:

- To make the benchmark easily accessible to the community, it would be necessary to share the code online.
- I know the page limit is tight, but I would have appreciated more details in the main paper.

Recommendation:

Overall, I think this is a good contribution to the AABI 2024 Workshop track.

---

### Official Review · Reviewer_Np24 · 2024-04-23
**Novel approach for benchmark tasks of estimating mutual information**

**Rating:** 7
**Confidence:** 4

**Review:**

This paper considers a benchmark task to test mutual information estimators and proposes a framework called Bend and Mix models (BMMs). It shows several theoretical aspects of the models and also shows how BMMs can be used to evaluate estimators and compute mutual information in the presence of outliers. It considers model-based Bayesian estimation of mutual estimation as well.

It tries to overcome the limitations of previous benchmark tasks such as Czyz et al. (2023) by introducing an approach to obtain new pointwise mutual information profiles based on a mixture of distributions and Monte Carlo sampling. It looks to be novel and applicable to several toy examples. However, the experimental studies dealt with by this paper is pretty simple, and it is still curious about if it could be used for real world problems, particularly high-dimensional problems where Monte Carlo sampling has difficulties to deal with, or problems related to long-tailed distributiones. In addition, it might be better to include more details about Bayesian inference-related examples considering the characteristics of this symposium.

Overall, the paper deals with a quite novel approach for a fundamental problem in machine learning and statistics by considering an example related to this symposium.

---

### Official Review · Reviewer_bpbG · 2024-04-25

**Rating:** 5
**Confidence:** 3

**Review:**

This paper introduces a few concepts.  The first is the "pointwise mutual information profile" which is the distribution of the random variable $\log p(X, Y) - \log p(X) -\log p(Y)$.  The expected value of this is the usual mutual information. The paper then considers classes of distributions (Bend and Mix Models) that can be obtained by taking a simple distribution on $X, Y$, transforming $X$ and $Y$ via diffeomorphisms, and then taking mixtures of such distributions.  The paper uses these distributions to benchmark different mutual information estimation methods, and proposes a Bayesian estimator of mutual information based on fitting this class of distributions to samples.

Overall, I found the idea of the PMI profile to be intriguing and there are some nice results in the paper, but overall the major contributions were unclear (see more detailed comments below) and the paper was often difficult to follow.

- One of the major motivations for introducing the PMI profile was that transformations used by recent benchmarks cannot alter the PMI profile.  I find this somewhat circular.  It would have been nice to provide additional intuition and theoretical results on what the higher moments of the PMI profile (beyond just the mean, which correspond to MI) mean and/or why they're important and necessary to infer accurately.  In general, I felt that the PMI profile was under-motivated.
- Some of the proofs in the appendix were difficult to follow.  In particular, some of the proofs seemed to prove much more than the stated result (e.g., the proof of proposition 8 continues proving additional results for several lines after proving the claim made in proposition 8).
- Beyond this, some roadmap of the results in the appendix (what they mean, why they're important, how they relate to the main text) would have been very helpful.  The current version feels like a compendium of theoretical results without any motivation or structure.
- In the introduction the statement "transforming simple variables cannot yield arbitrary continuous distributions" should be rephrased to be more precise.  Presumably this is referring to marginal transformations being unable to produce arbitrary continuous joint distributions.  Otherwise, the inverse CDF trick transforms uniform r.v.s. t o arbitrary continuous distributions.
- The acronym "PMI" is not defined prior to its first mention in "contributions".

---

### Official Review · Reviewer_gnw2 · 2024-04-25

**Rating:** 7
**Confidence:** 3

**Review:**

In this paper, the authors studied pointwise mutual information profiles, and proposed Ben and Mix Models (BMMs), the profile of which can be used to provide multiple use cases for uncertainty quantification in Bayesian inference.

In terms of writing, this paper needs to include an abstract section as the template requires. To my best knowledge, the math derivations in this paper are sound. On the other hand, I suggest that the authors can provide a more comprehensive literature review, because I personally am confused about how this work aligns in the corresponding line of work.

---

### Official Review · Reviewer_Jsc7 · 2024-04-26

**Rating:** 6
**Confidence:** 2

**Review:**

The paper studies the estimation of pointwise mutual information (PMI) and introduces the Bend and Mix Models (BMMs), which enable the approximation of arbitrary distributions for robust mutual information estimation. Using BMMs, the study addresses limitations of existing benchmarks by creating more versatile and challenging tasks. Authors demonstrate its usefulness through  empirical case study.

Strengthens:
1. Introducing BMMs adds a significant advancement in the field of mutual information estimation by allowing for the approximation of arbitrary distributions.
2. Comprehensive Theoretical insights

Weaknesses:
1. More baselines are preferred. How does BMM compare with CKA?
2. Domain Knowledge is required.

---

### Official Review · Reviewer_T4HU · 2024-04-26
**A comprehensive contribution on Pointwise Mutual Information estimation**

**Rating:** 6
**Confidence:** 3

**Review:**

# Summary Of Contributions
This work critiques the task generation method by Czyz et al. (2023), demonstrating its inability to model arbitrary distributions and showing that marginal transformations do not alter the PMI profile, with an analytical determination of this profile for multivariate normal distributions. It introduces Bend and Mix Models (BMMs), which allow for reliable estimation of true mutual information (MI) using Monte Carlo sampling and presents three applications: approximating arbitrary distributions, assessing the robustness of MI estimators, and estimating MI with epistemic uncertainty using a parametric model, thereby generalizing the CCA estimator.

I need to stress that I did not check the Appendix, which is quite long (20 pages).

# Strengths And Weaknesses
### Strengths
- Estimating PMI accurately is crucial given its broad applicability across various fields.
- I like the idea of introducing a flexible class of benchmark.
- The material is presented clearly and effectively.
- This is a quite comprehensive work for a workshop paper, with a lot of extra material in appendix. Skimming through this appendix, it seems solid, but this is only an educational guess.

### Weaknesses
- Add a summary!! I don't have much to critique here, probably due to not being expert enough in mutual information.

---

### Official Review · Reviewer_4MR9 · 2024-04-26

**Rating:** 8
**Confidence:** 4

**Review:**

The paper presents a comprehensive study on mutual information (MI) estimation techniques with a focus on continuous random variables (r.v.s). The authors introduce a novel class of distributions, referred to as Bend and Mix Models (BMMs).
Key contributions of the work include:
- A critique of existing mutual information benchmarks (Czyz et al. ˙(2023)), highlighting their inability to model arbitrary continuous distributions (Theorem 2).
- The introduction of Bend and Mix Models (BMMs) that combine trough mixtures several nonlinear transformation of random variables. Even if the PMI profile is not analytically known, it can be easily estimated via MonteCarlo sampling.

On the experimental side, the authors show that:
- Existing MI estimators (MINE, InfoNCE, etc...) fails on the distributions which are outside the Czyz et al. ˙(2023) benchmark. In particular in Figure 2 it is shown how on challenging tasks (e.g. Waves) all considered estimators fail.
- The BMM models can be used to analyze systems with experimental failures. In particular, the authors investigate the different behaviour of inlier and outlier noise (Figure 3).
- The BMM models can be used as a building block for model-based mutual information estimators, as for example the results in Figure 4 where mixture of Gaussians are used to estimate the posterior distrubution of MI given observation samples.

Some of the (minor) weaknesses or open points of the work are:
- Motivation: the authors could better justify why the PMI profile is such an important quantity. It is well understood that diffeomorphism cannot change MI or PMI profiles, but why is changing the whole PMI profile an important problem? Why are existing estimators unreliable on distributions from the bend and mix models?
- Empirical Validation: Although the paper presents some empirical results, the testing on real-world data seems limited. More extensive empirical evaluations across diverse real-world datasets could strengthen the findings. Moreover, the authors could include other classes of MI estimators (e.g. diffusion models based [1] or normalizing flows based [2])

[1] Franzese et a. "MINDE: Mutual Information Neural Diffusion Estimation"  ICLR 2024, https://openreview.net/pdf?id=0kWd8SJq8d
[2] Butakov et al. "Mutual Information Estimation via Normalizing Flows" https://arxiv.org/pdf/2403.02187

---

### Meta-Review · Area_Chair_oL1B · 2024-05-13

**Recommendation:** Accept (Poster)
**Confidence:** 4

**Metareview:**

This paper identifies issues in the existing mutual information estimation methods and proposes a method to improve estimation accuracy.

---

### Decision · Program_Chairs · 2024-05-27

Accept